# Spectral Multiple-Instance Learning for Efficient Gigapixel Image Analysis

## Abstract

With ongoing advances in imaging technology, gigapixel images are now widely utilized in both scientific research and industrial applications. However, their extremely large scale presents significant challenges for conventional deep learning workflows. A common approach involves partitioning the image into thousands of smaller patches, processing each patch independently, and aggregating the representations using a Multiple-Instance Learning (MIL) framework. Because the label of a gigapixel image often depends on a small subset of informative regions, identifying these key patches is essential. However, MIL faces a persistent multi-resolution dilemma: low-magnification views offer global contextual information but fail to capture fine-grained details, whereas high-magnification views retain these details at a substantial computational cost. We introduce Multi-Instance Learning with Spectral Methods (SpecMIL), which addresses this challenge by capturing high-frequency features at low magnification and preserving geometric relationships across scales using graph spectral theory. SpecMIL exploits spectral features that remain informative even after down-sampling, guiding selective high-resolution "zoom-in" only where necessary. Experiments on various whole slide image benchmarks (e.g., tumor subtyping, grading, and metastasis detection) demonstrate that spectral approaches offer a highly effective and efficient solution for gigapixel image analysis.

## 1 Introduction

In recent years, the demand for processing high-resolution data has grown markedly across multiple areas of computer vision, including image generation Wu et al. (2025), Vision–Language Models (VLMs) Shi et al. (2024), large-scale scene understanding Ma et al. (2024), and medical image analysis Kapse et al. (2024). A prominent example is the analysis of whole-slide images (WSIs) Lu et al. (2021), whose gigapixel resolution can reach up to $150,000 \times 150,000$ pixels Zhang et al. (2022). Such sheer size renders WSIs impractical to process with standard vision encoders under current hardware constraints, creating bottlenecks for downstream tasks such as disease diagnosis, tumor grading, and tissue classification.

A pragmatic workaround is the multiple-instance learning (MIL) pipeline, which partitions a gigapixel image into smaller, computationally manageable patches. Each patch is encoded independently, and the resulting embeddings are aggregated to form a slide-level representation. Despite its practicality, this pipeline faces a fundamental challenge: the inherent trade-off between global context and local resolution. Low-resolution views capture global tissue organization but miss fine-grained morphology, whereas high-resolution patches reveal cellular details but lose broader context. Capturing both simultaneously is therefore difficult.

This multi-resolution dilemma is not unique to WSIs; it arises in any gigapixel-scale image domain. Balancing the preservation of global structure against the resolution needed for detailed analysis is a pervasive problem across high-resolution imaging applications.

Prior work tackles the issue with hierarchical transformers Chen et al. (2022); Guo et al. (2023) or by selectively sampling a small set of high-magnification patches guided by low-magnification cues Thandiackal et al. (2022). While these strategies improve accuracy, they incur substantial computational overhead or rely on identifying subtle signals that may vanish at low magnification. Moreover, modern vision encoders that achieve state-of-the-art performance on benchmark datasets

have a strong tendency to *smooth out* high-frequency features, making it even harder to retain fine morphological cues in low-resolution views. Consequently, important signals become nearly invisible at low magnification, whereas their corresponding high-magnification patches are prohibitively expensive to enumerate exhaustively.

To overcome this dilemma, we introduce **Multi-Instance Learning with Spectral Methods (SpecMIL)**, a selective-magnification framework that harnesses spectral methods to preserve high-frequency morphological cues while remaining computationally efficient. As detailed in Section 4, SpecMIL operates in two steps: (1) it extracts high-frequency features from each low-magnification patch, and (2) it encodes rotation-invariant geometric relationships among patches by constructing a graph and applying a novel *Learnable Geometric Position Encoding* (LGPE). By detecting subtle signals that usually require higher magnification, SpecMIL "zooms in" only on the most relevant regions, dramatically reducing the explosion in patch count at higher magnifications. The patch-graph edges capture both spatial proximity and similarity in high-frequency feature space, while LGPE leverages graph spectral analysis to make the positional encoding robust to arbitrary slide rotations.

In this way, SpecMIL preserves diagnostic features that are otherwise lost at low magnification and avoids the heavy cost of scanning an entire slide at maximum magnification. It also removes the need for the differentiable top-$k$ operation used in prior zoom-based MIL frameworks Thandiackal et al. (2022), thereby lowering memory usage and computation. Together, these spectral- and geometry-driven design choices enable SpecMIL to resolve the multi-resolution dilemma and to achieve state-of-the-art accuracy with markedly improved efficiency, as demonstrated in our experiments.

## 2 RELATED WORK

**Multiple Instance Learning.** MIL has emerged as the dominant paradigm for *any* gigapixel-scale image, where only image-level supervision is attainable and loading the full resolution would overwhelm GPU memory. Early explicit MIL pipelines fused patch-level scores through mean or max pooling Campanella et al. (2019); Zhang et al. (2022), but attention–based aggregation such as ABMIL Ilse et al. (2018) quickly became the standard because it learns instance weights end-to-end and yields interpretable region-importance maps. To improve the discriminative power of patch representations, CLAM Lu et al. (2021) introduces an auxiliary task within the MIL framework to evaluate instance relevance based on attention size for WSI classification. TransMIL Shao et al. (2021) captures relationships among patches using self-attention Vaswani et al. (2017), addressing the computational complexity of softmax in long sequences through linear Self-Attention Wang et al. (2020b); Xiong et al. (2021).

**Multi-resolution WSIs.** To leverage the rich information offered by multiple magnifications, DSMIL Li et al. (2021) embeds multi-scale representations by concatenating patch features across different scales. H2-MIL Hou et al. (2022) represents WSIs as hierarchical heterogeneous graphs, whereas HIPT Chen et al. (2022) introduces a hierarchical pre-training framework specifically designed for multi-resolution analysis. HIGT Guo et al. (2023) further expands upon these ideas by incorporating a novel Bidirectional Interaction module that captures both local and global information from WSI pyramids simultaneously. Despite these advances, the above methods primarily emphasize either local or global correlations within WSI pyramids, limiting their capacity to fully exploit complex, multi-resolution information. ZoomMIL Thandiackal et al. (2022), our main baseline, proposes a "zoom" process that selects key regions at low magnification and retrieves corresponding patches at higher magnification, effectively

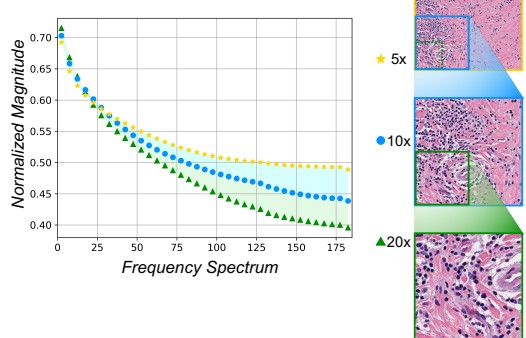

Figure 1: Normalized magnitude of the frequency spectrum. Lower magnification images exhibit higher magnitudes in the high-frequency spectrum compared to higher magnification images (left). When visual features at high magnification images are compressed into smaller spatial areas at lower magnification (right), the features are shifted into higher frequency components within the frequency domain.

mirroring a pathologist's diagnostic workflow. Nonetheless, ZoomMIL still relies on a differentiable top-$k$ operation, which can be computationally expensive and risks missing fine morphological cues if they are not adequately represented at low magnification.

**High-Frequency Components in WSIs.**   In the context of WSIs and digital pathology, understanding the relationship between magnification and frequency components is crucial for accurate image analysis and interpretation. High-frequency components in an image correspond to fine details and sharp transitions, such as cellular structures and tissue boundaries, which are often critical for diagnostic purposes. The morphology observed at high magnification corresponds to high-frequency components when viewed at lower magnifications. This occurs because the same visual patterns that occupy a larger spatial area at high magnification are compressed into a smaller spatial area in the lower-magnification image, as illustrated on the right side of Fig. 1. As a result, the fine details and sharp transitions inherent in these high-magnification features are translated into higher frequency components within the frequency domain of the lower-magnification image. Understanding this relationship is essential for ensuring the fidelity of morphological analysis across different levels of magnification in WSIs. We provide a theoretical explanation of the relationship between frequency and resolution in the appendix.

**Learnable Structural and Positional Encoding.**   Positional encoding is a fundamental component in modern deep learning models, particularly Transformer architectures, as it enables the model to capture the order of elements within a sequence. This capability is essential for understanding contextual relationships and maintaining sequence information in tasks like natural language processing. Such encoding is crucial for allowing the model to learn patterns based on both content and position, thereby improving performance in tasks where sequence order plays a significant role. LSPE Dwivedi et al. (2021) has been proposed to enhance the representational power of positional encoding by learning position representations separately, rather than merging them directly with node features. However, LSPE treats the positional encoding topology as identical to the message-passing graph topology. In our approach, we propose a refined positional encoding strategy that leverages spectral positional encoding derived from both distance-based adjacency and feature-based adjacency for geometric connectivity.

## 3   PRELIMINARY

In this preliminary, we start with a basic MIL pipeline for WSIs. Then, we explain the basic concepts of Graph Neural Network (GNN) architecture that is utilized to incorporate inherent geometry of MIL patches in our proposed Learnable Geometric Position Encoding (LGPE).

**Multi-Instance Learning for WSI.**   Given a WSI $X$, the slide-level prediction $\hat{Y}$ is obtained by learning a classifier $f(X; \theta)$. Due to the extensive resolution of WSIs, $X$ is patched into a bag of small instances $X = \{x_1, ..., x_N\}$, where $N$ is the number of instances. Then, the slide-level prediction $\hat{Y}$ is obtained by a global-pooling operation of the latent label $\hat{y}_i$ for each instance $x_i$, which can be defined as:

$$\hat{Y} = \max\{\hat{y}_1, ..., \hat{y}_N\}. \tag{1}$$

Since there are no labels for each instance $\hat{y}_i$ under WSI-level supervision, conventional approaches convert this problem into a MIL formulation with two steps: 1) Processing images into feature representations $Z = \{z_1, ..., z_N\}$ with a backbone $f_v$ as $z_i = f_v(x_i; \theta_1), z_i \in \mathbb{R}^C$ where $f_v$ is a model of any architecture such as CNN or ViT with parameters $\theta_1$. 2) Aggregating the features of all patches within a slide and producing the slide-level prediction $Y = g(Z; \theta_2)$, where $g$ is an attention-based pooling function followed by a linear classifier head as:

$$g(Z; \theta_2) = \sigma\big(\sum_{i=1}^{N} a_i z_i\big), \tag{2}$$

where $a_i$ is attention weights and $\sigma(\cdot)$ is a linear head. Limited by computational cost, the parameters $\theta_1$ and $\theta_2$ in $f(X; \theta) = g\{f_v(X; \theta_1); \theta_2\}$ are learned separately by the following steps: 1) Initializing $\theta_1$ from the pretrained model. 2) Freezing $\theta_1$ and learning $\theta_2$ under slide-level supervision.

Figure 2: Overall architecture of SpecMIL.

**WSIs as graph.** Let $\mathcal{G} = (\mathcal{V}, \mathcal{E})$ be a graph with $\mathcal{V}$ being the set of nodes and $\mathcal{E}$ the set of edges. Each WSI image $X$ and its patches can be represented as $\mathcal{G}$ and $\mathcal{V}$, respectively. The graph has $N = |\mathcal{V}|$ nodes and $E = |\mathcal{E}|$ edges. The connectivity of the graph is represented by the adjacency matrix $A \in \mathbb{R}^{N \times N}$ where $A_{ij} = 1$ if there exists an edge between the nodes $i$ and $j$; otherwise $A_{ij} = 0$. The degree matrix is denoted $D \in \mathbb{R}^{N \times N}$. The update equation for the node representaion $z_i$ with a conventional GNN layer is defined as:

$$z_i^{\ell+1} = f_{\text{GNN}}(z_i^\ell, \{z_j^\ell\}_{j \in \mathcal{N}_i}), \quad z_i^{\ell+1}, z_i^\ell \in \mathbb{R}^C, \tag{3}$$

where $f_{\text{GNN}}$ is a function with learnable parameters, and $\mathcal{N}_i$ is the neighbor nodes of the node $i$.

## 4 METHOD

Here, we present SpecMIL, a selective magnification strategy designed to tackle the *multi-resolution dilemma* of WSIs, specifically addressing the trade-off between information richness and complexity at different magnification levels. SpecMIL employs two spectral methods for informative features: 1) Frequency analysis to extract high-frequency components from images, and 2) Graph spectral analysis to capture and encode the geometric properties between patches. By selecting regions to *zoom* based on high-frequency components and geometric positional encoding at low-magnification patches, SpecMIL successfully mitigates the multi-resolution dilemma of WSIs, accurately capturing morphological features that require higher magnification exploration at low magnification levels. Through this hierarchical magnification selection architecture, SpecMIL significantly diminishes the complexity of exploring high-magnification in a quadratic magnitude while improving performance. The overall architecture of SpecMIL and our proposed **L**earnable **G**eometric **P**osition **E**ncoding (LGPE) are illustrated in Fig 2 and Fig 3, respectively.

### 4.1 MULTI-RESOLUTION PATCH ENCODING

To begin, we crop the WSI $X$ into multiple non-overlapping image patches under different magnifications (i.e., $\times 5$, $\times 10$) with sliding windows. OTSU filtering is applied to filter out the background patches. Then, we encode the original image patches for low magnification image as set $X_{\text{low}}$ and patches from $M$ times higher magnification as set $X_{\text{high}}$. Each $X_{\text{low}}$ and $X_{\text{high}}$ is written as:

$$X_{\text{low}} = \{x_1, x_2, \cdots, x_N\}, \quad X_{\text{high}} = \{\bar{x}_{11}, \bar{x}_{12}, \cdots, \bar{x}_{1M^2}, \cdots, \bar{x}_{N1}, \bar{x}_{N2}, \cdots, \bar{x}_{NM^2}\}, \tag{4}$$

where $N$ is the total number of patches at low magnification, high magnification patches $\{\bar{x}_{i1}, \bar{x}_{i2}, \cdots, \bar{x}_{iM^2}\}_{i=1,2,\cdots,N}$ corresponds to the low magnification region $x_i$ since each patch generates $M^2$ patches at $M$ times higher magnification. Then, the patches are encoded with a vision encoder $f_v$ (e.g., ResNet50), resulting in features $Z_{\text{low}} = \{z_1, z_2, \cdots, z_N\}$ and $Z_{\text{high}} = \{\bar{z}_{11}, \bar{z}_{12}, \cdots, \bar{z}_{1M^2}, \cdots, \bar{z}_{N1}, \bar{z}_{N2}, \cdots, \bar{z}_{NM^2}\}$ where $z_i, \bar{z}_{ij} \in \mathbb{R}^{1 \times C}$.

## 4.2 SPECTRAL HIGH-FREQUENCY ENCODING

Given a patch image $x$, Fourier transform $\mathcal{F}(\cdot)$, the high frequency component $\tilde{x}^h$ is written as:

$$\tilde{x} = \mathcal{F}(x), \tilde{x}^h = t(\tilde{x}; r), \tag{5}$$

where $t(\cdot; r)$ denotes the channel-wise thresholding function Wang et al. (2020a) that separates the high frequency components from $\tilde{x}$ according to a hyperparameter radius $r$. Then, the final high-frequency encoding from patch $x$ becomes $z^h = f_v(\mathcal{F}^{-1}(\tilde{x}^h))$ where $\mathcal{F}^{-1}(\cdot)$ denotes inverse Fourier transform.

## 4.3 LEARNABLE GEOMETRIC POSITION ENCODING

We propose a positional encoding based on spectral graph theory and adjacency matrix calculated by high-frequency features. LGPE is invariant to slide rotation, accurately encoding positional information of patches in WSIs. By constructing a graph where image patches serve as nodes and their global connectivity or geometric proximity as edges, LGPE is designed to encode inherent geometry using *distance*-based adjacency $A_{\text{dis}}$ and capture geometry within the tissue using the *feature*-based adjacency $A_{\text{HF}}$, derived from the similarity of high-frequency features (e.g., patterns of fat cells, cytoplasm, and metastasis).

**Distance-based Geometry.** The distance-based adjacency $A_{\text{dis}}$ is constructed by $k$-NN on Euclidean distance between patches with distance thresholding applied (thus, $A_{\text{dis}}(i, j) = 1$ if $j$-th patch is within the top-$k$ nearest patches of $i$-th patch and the distance between them does not exceed the threshold; otherwise, $A_{\text{dis}}(i, j) = 0$). The positional encoding (i.e., Laplacian eigenvectors) is then obtained by $A_{\text{dis}}$ based on spectral techniques Belkin & Niyogi (2003). The graph Laplacian is decomposed as:

$$\Delta = I_n - D_{\text{dis}}^{-1/2} A_{\text{dis}} D_{\text{dis}}^{-1/2} = U^T \Lambda U, \tag{6}$$

where $I_n$, $D_{\text{dis}}$, $\Lambda$, and $U$ are the identity matrix, the degree matrix, the eigenvalues, and the eigenvectors, respectively, all in $\mathbb{R}^{N \times N}$. The eigenvectors are sorted by the magnitude of corresponding eigenvalues. Since these eigenvectors are calculated from $A_{\text{dis}}$ based on the Euclidean distance between patches, the positional

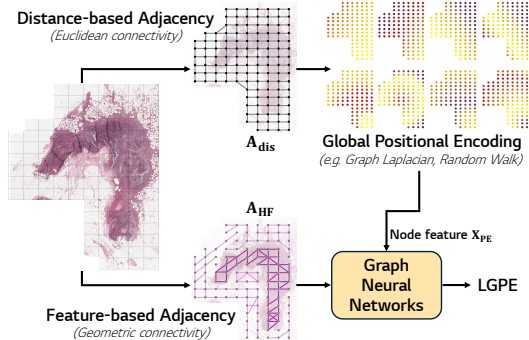

Figure 3: Illustration of LGPE. Visualizations of $A_{\text{dis}}$, $A_{\text{HF}}$, and $X_{\text{PE}}$ are included in the appendix. Additionally, a comparison between $A_{\text{HF}}$ and $A_{\text{LF}}$ (i.e. adjacency based on low-frequency features) is provided in the appendix to support the rationale for using high-frequency feature similarity.

encoding holds key properties such as invariance to slide rotation, which is crucial for WSIs.

**HF Feature-based Geometry.** To capture the inherent geometry (formed by cells, cytoplasm, and glandular structures) that extends beyond patch-level scopes or Euclidean distances, we utilize a similarity matrix of high-frequency features encoded by our backbone $f_v$. Given the similarity matrix $S \in \mathbb{R}^{N \times N}$, where $S_{i,j}$ is the similarity (e.g., cosine similarity) between patches $h_i$ and $h_j$, the connectivity is defined as:

$$e_{i,j} = \begin{cases} 1, & \text{if } d(i,j) \leq \tau \text{ and } j \in \text{top-}k(S_{i,:}) \\ 0, & \text{otherwise} \end{cases} \tag{7}$$

where $\tau$ is a distance threshold, and top-$k(\cdot)$ is the function that returns the indices of the top $k$ similarity patches. Subsequently, we construct a symmetric adjacency matrix $A_{HF}$.

**LGPE: Combined Geometry.** Based on the global eigenvectors $U$ from Eq.(6) and the adjacency matrix $A_{\text{HF}}$, LGPE is obtained with:

$$X_{PE} = \tanh(U_{[:,:k_{PE}]}W), \; z^p = \text{GNN}(X_{PE}, A_{\text{HF}}), \tag{8}$$

where $\tanh(\cdot)$ is the hyperbolic tangent function, $\text{GNN}(\cdot)$ is a conventional GNN at Eq.(3), and $k_{PE}$ is the number of selected eigenvectors. Additionally, other graph positional encodings, such as Random Walk Li et al. (2020), can be used as $X_{PE}$. Empirical results show that geometric connectivity (or feature-based adjacency) with high-frequency components successfully encodes meaningful inherent geometry, positively contributing to the final performance (see Table 4).

**Invariance to Slide-level Rotation.** A critical property of positional encoding in the context of WSIs is its invariance to slide-level rotations. Since WSIs typically lack a consistent orientation—meaning there is no predefined "top," "bottom," "left," or "right"—it is essential that the positional encoding scheme remains robust to such variations. This invariance ensures that the encoded positions of pixels or regions within the slide are not affected by arbitrary rotations of the image, which is a common occurrence in histopathological analysis. Consequently, the positional encoding must be designed in a manner that it provides a consistent and orientation-independent representation of spatial information. When constructing the global positional encoding from the distance-based adjacency matrix $A_{\text{dis}}$, it satisfies rotation invariance as demonstrated in the following proposition. The proof is provided in the appendix.

**Proposition 1** (Rotation Invariance of the Adjacency Matrix Derived from Euclidean Distances). *Let $P_1, P_2, \ldots, P_n$ be the coordinates of $n$ patches in a plane, and let $d(P_i, P_j)$ denote the Euclidean distance between patches $P_i$ and $P_j$. Define the adjacency matrix $A$ such that the $(i,j)$-th entry $A_{ij} = d(P_i, P_j)$. If the slide is rotated by an angle $\theta$ around a point $O$, then the resulting adjacency matrix $A'$ after rotation is identical to $A$, i.e., $A' = A$.*

**LGPE-aware Gated Attention.** **L**GPE-aware **G**ated **A**ttention (LPGA) is a modified gated attention, which is widely used in attention-based MIL methods Ilse et al. (2018). To aggregate patch-level features, the attention coefficients of LPGA are calculated as:

$$\alpha_i = \frac{\exp(w^\top(\tanh(V z_i^{PE}) \odot \eta(U z_i^{PE})))}{\sum_{j=1}^N \exp(w^\top(\tanh(V z_j^{PE}) \odot \eta(U z_j^{PE})))}, \tag{9}$$

where $z_i^{PE} = [z_i; z_i^h; z_i^p]$, $w \in \mathbb{R}^{L \times 1}, V \in \mathbb{R}^{L \times D}, U \in \mathbb{R}^{L \times D}$ are learnable parameters, $\eta(\cdot)$ is the sigmoid function, and $\odot$ is element-wise multiplication. For selected patches at higher magnification, we use the LGPE corresponding to the lower-magnification (further referred to as Aligned LGPE). Consequently, the input for LPGA at higher magnification is computed as $\bar{z}_{ij}^{PE} = [\bar{z}_{ij}; \bar{z}_{ij}^h; z_i^p]$. By aligning LGPE between identical regions across different magnifications, SpecMIL ensures that consistent geometric information are applied to heterogeneous patches while eliminating the need to compute positional encoding on the huge amount of high magnification patches.

## 4.4 Zooming and Feature Aggregation

**Zooming of SpecMIL.** SpecMIL addresses the multi-resolution dilemma in WSI analysis by leveraging high-frequency domain components to determine where and when to *zoom in*. This enables SpecMIL to discriminate whether to *zoom* or not at a lower magnification, serving more detailed visual patterns for higher magnifications at a lower level.

**Multi-magnification Aggregation.** SpecMIL overcomes the dilemma of multi-resolution by relying the zooming process on high-frequency features. We found that the features from different magnifications result in very heterogeneous features. Therefore, instead of dealing with multiple magnification features with a single network, SpecMIL features multiple networks, each of which deals with its own magnification, then sums the results at the late layer.

$$g(Z^c; \theta_2) = \sigma\Big(\sum_{i=1}^N \alpha_i z_i^c\Big), \quad \bar{g}(\bar{Z}^c; \theta_3) = \sigma\Big(\sum_{i=1}^N \sum_{j=1}^{M^2} \mathbf{T}_j \alpha_{ij} \bar{z}_{ij}^c\Big), \tag{10}$$

where $\mathbf{T} \in \{0,1\}^{M^2}$ denotes the indicator matrix of the regions at high magnification corresponding to the top-$k$ region selected at low magnification. The final slide-level prediction is calculated as $\hat{Y} = f(g(Z^c), \bar{g}(\bar{Z}^c))$ where $f$ is a linear projection.

## 5 Experiments and Discussion

In this section, we evaluate the performance of the proposed method integrated with the latest WSI-MIL frameworks. For a fair comparison, we utilize ResNet50 as the fixed backbone $f_v$, with its frozen parameters $\theta_1$ pretrained on ImageNet 1K. It is important to note that our method is designed to be backbone-agnostic for MIL, meaning the backbone can be replaced without causing any conflicts.

## 5.1 DATASETS

We validate SpecMIL on three H&E stained, public WSI datasets. Following ZoomMIL Thandiackal et al. (2022), SpecMIL utilizes all patches at low magnification and selects $k \times M^2$ patches at high magnification, where $k$ represents the number of selected patches. For reference of how efficient our proposed SpecMIL is, we also show the average number of patches for each magnification and each dataset.

**CAMELYON16.** The CAMELYON16 dataset comprises 270 WSIs for training, including 160 normal slides and 110 with metastases, along with 129 slides designated for testing. These slides were scanned at 40x magnification using 3DHISTECH and Hamamatsu scanners at the Radboud University Medical Center and the University Medical Center Utrecht in the Netherlands. For our study, we stratified the 270 training slides into a 90%/10% split for training and validation purposes. On average, each image contains 2,472 patches at $10\times$ magnification and 9,887 patches at $20\times$ magnification.

**BRIGHT.** The BRIGHT dataset includes WSIs of breast tissue, covering non-cancerous, precancerous, and cancerous subtypes. These slides were collected at the Fondazione G. Pascale in Italy and digitized using an Aperio AT2 scanner at 40x magnification. For our work, we utilized the BRIGHT challenge splits, which provide 423 WSIs for training, 80 for validation, and 200 for testing. Each image contains an average of 61 and 246 patches at $1.25\times$ and $2.5\times$ magnifications, respectively.

**CRC.** The CRC dataset comprises 1,133 digitized colorectal biopsy and polypectomy slides, categorized into nonneoplastic, low-grade, and high-grade lesions, which make up 26.5%, 48.7%, and 24.8% of the data, respectively. These slides were collected at the IMP Diagnostics laboratory in Portugal and digitized using a Leica GT450 scanner at

| Methods | Weighted-F1 (%) | Acc (%) |
|---|---|---|
| MAXMIL (20×) | 68.9 ±9.0 | 74.7 ±3.6 |
| MEANMIL (20×) | 60.0 ±2.0 | 62.3 ±2.9 |
| ABMIL (20×) | 84.5 ±1.9 | 84.8 ±2.0 |
| CLAM-SB (20×) | 83.3 ±3.1 | 83.6 ±3.0 |
| TRANSMIL (20×) | 77.9 ±4.9 | 78.6 ±4.9 |
| DSMIL-LC (10× + 20×) | 81.1 ±2.3 | 81.5 ±2.3 |
| $R^2$T-MIL (10×) | 76.7 ±1.9 | 77.5 ±2.1 |
| $R^2$T-MIL (20×) | 80.3 ±5.1 | 80.6 ±5.2 |
| $R^2$T-MIL (10× + 20×) | 77.1 ±6.0 | 78.1 ±6.0 |
| WiKG (10×) | 77.6 ±2.6 | 78.9 ±2.3 |
| WiKG (20×) | 86.9 ±2.7 | 87.3 ±2.7 |
| WiKG (10× + 20×) | 77.0 ±1.5 | 78.6 ±1.6 |
| ZoomMIL (10× → 20×) | 77.7 ±2.5 | 78.7 ±2.3 |
| **SpecMIL** (10× → 20×) | **87.0** ±1.3 | **87.5** ±1.2 |

Table 1: Performance comparison on the CAMELYON16 dataset. The symbol '+' indicates that the method incorporates features from both magnifications, while the symbol '→' represents the zooming process. Extended results, including the multi-resolution adaptations of all baselines, are provided in the appendix. These results highlight the non-trivial challenges of leveraging multi-resolution approaches. Note that the reported performance may differ from previously published results, as we conduct experiments using a greater variety of data splits and random seeds to ensure robustness.

| Methods | Weighted-F1 (%) | Acc (%) |
|---|---|---|
| MAXMIL (2.5×) | 64.9 ±3.9 | 65.5 ±4.6 |
| MEANMIL (2.5×) | 63.6 ±4.3 | 63.4 ±4.4 |
| ABMIL (2.5×) | 70.0 ±1.6 | 69.9 ±1.8 |
| CLAM-SB (2.5×) | 68.0 ±3.2 | 68.3 ±3.6 |
| TRANSMIL (2.5×) | 65.2 ±4.6 | 66.3 ±5.4 |
| DSMIL-LC (1.25× + 2.5×) | 67.1 ±1.3 | 68.2 ±2.6 |
| $R^2$T-MIL (1.25×) | 62.9 ±2.0 | 63.4 ±2.4 |
| $R^2$T-MIL (2.5×) | 69.6 ±2.2 | 69.8 ±2.9 |
| $R^2$T-MIL (1.25× + 2.5×) | 68.1 ±2.3 | 68.9 ±3.0 |
| WiKG (1.25×) | 65.9 ±2.3 | 66.0 ±2.7 |
| WiKG (2.5×) | 68.8 ±2.0 | 69.0 ±2.9 |
| WiKG (1.25× + 2.5×) | 66.5 ±2.1 | 66.3 ±2.7 |
| ZoomMIL (1.25× → 2.5×) | 66.5 ±2.8 | 66.4 ±3.1 |
| **SpecMIL** (1.25× → 2.5×) | **70.7** ±4.7 | **71.7** ±4.8 |

Table 2: Performance on BRIGHT dataset.

40x magnification. The average number of patches at $5\times$ and $10\times$ magnifications is 213 and 853, respectively.

## 5.2 BASELINES

We compare SpecMIL with MIL methods in literature. Specifically, we compare with ABMIL, which uses a gated-attention pooling, and its variant CLAM, which also includes an instance-level clustering loss. We further compare with two spatially-aware methods, namely, TransMIL which models instance-level dependencies using transformer-based pooling. In addition, we compare with multi-scale methods DSMIL, which is a multi-magnification approach that encodes all patches in a WSI across all considered magnifications. For completeness, we also include vanilla MIL

| Methods | Weighted-F1 (%) | Acc (%) |
|---|---|---|
| MAXMIL ($10\times$) | 87.9 ±1.9 | 87.8 ±2.1 |
| MEANMIL ($10\times$) | 87.3 ±0.9 | 87.3 ±0.9 |
| ABMIL ($10\times$) | 90.3 ±1.7 | 90.2 ±1.8 |
| CLAM-SB ($10\times$) | 91.6 ±1.3 | 91.6 ±1.4 |
| TRANSMIL ($10\times$) | 90.4 ±2.5 | 90.3 ±2.7 |
| DSMIL-LC ($5\times + 10\times$) | 90.6 ±1.7 | 90.6 ±1.8 |
| $R^2$T-MIL ($5\times$) | 90.4 ±1.4 | 90.4 ±1.4 |
| $R^2$T-MIL ($10\times$) | 91.4 ±0.9 | 91.4 ±0.9 |
| $R^2$T-MIL ($5\times + 10\times$) | 92.3 ±1.3 | 92.3 ±1.3 |
| WiKG ($5\times$) | 88.7 ±1.8 | 88.7 ±1.8 |
| WiKG ($10\times$) | 91.9 ±1.4 | 91.9 ±1.4 |
| WiKG ($5\times + 10\times$) | 91.7 ±2.1 | 91.6 ±2.2 |
| ZoomMIL ($5\times \rightarrow 10\times$) | 91.6 ±1.1 | 91.6 ±1.1 |
| **SpecMIL** ($5\times \rightarrow 10\times$) | **92.3** ±0.8 | **92.4** ±0.8 |

Table 3: Performance on CRC dataset.

methods such as MaxMIL and MeanMIL. $R^2$T-MIL Tang et al. (2024) enhances representations by re-embedding instance features with regional and cross-region attention. Recently, WiKG Li et al. (2024) has been proposed that utilizes a dynamic graph representation algorithm for WSIs. Note that SpecMIL operates efficiently with a small number of patches and low GPU usage (see Fig 4), indicating its potential for integrating additional techniques from existing literature, such as updating $z_i$ by their methodologies. ZoomMIL Thandiackal et al. (2022) is set as a notable baseline since it employs a strategy similar to SpecMIL, utilizing hierarchical patch selection from low to high magnification (e.g., $10\times \rightarrow 20\times$). At low magnification, patches are selected using a differentiable top-k approach Cordonnier et al. (2021) combined with a perturbed optimizer Berthet et al. (2020). Additional implementation details and hyper-parameters are provided in the supplemental material. For a fair comparison, we perform identical preprocessing, including the extraction of patch embeddings.

**Preprocessing.** For each WSI, we detect the tissue area using a Gaussian tissue detector and divide the tissue into $256 \times 256$ patches at all considered magnifications. Subsequently, we encode the patches into 1024-dimensional embeddings using the pre-trained ResNet50 $f_v$, with adaptive average pooling applied after the third residual block.

**Configuration.** The experiments in our paper are conducted on the code base of ZoomMIL on a single NVIDIA A100 GPU. For baselines, we use the reported best configurations including learning rate, scheduler, weight decay, number of epochs, and dropout probability. For SpecMIL, the number of eigenvectors $k_{PE}$ is set to 8, with the projection dimension defined as $2k_{PE}$. For CRC, BRIGHT, and CAMELYON16, we select 16, 20, and 600 patches at low magnification, respectively (denoted

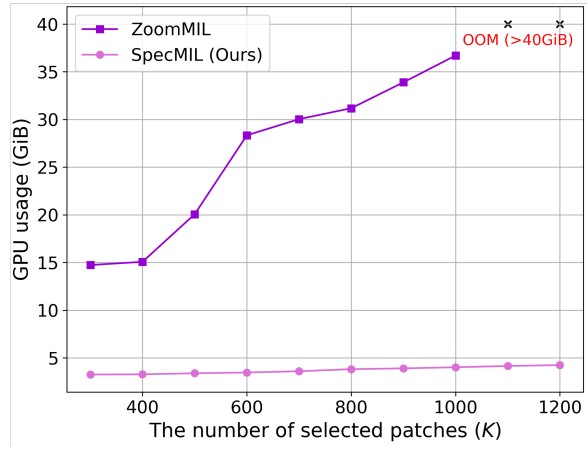

Figure 4: Comparison of GPU usage between SpecMIL and ZoomMIL, which both deploys hierarchical patch selection strategy from low to high magnification ($10\times \rightarrow 20\times$ on the CAMELYON16 dataset). Our approach, which integrates rich information (e.g., geometry and high-frequency features) at low magnification and removes the need for a complex top-k selection module, significantly reduces GPU memory usage. This efficiency allows for the potential integration of additional modules. Additional metrics, including FLOPs and computation time, are provided in the appendix.

as $K$ in Fig 4). Therefore, SpecMIL takes a maximum of 64, 80, and 2,400 patches at high magnification for CAMELYON16 ($20\times$), BRIGHT ($2.5\times$), and CRC ($10\times$), respectively. In constructing $A_{\text{dis}}$ and $A_{\text{HF}}$, the distance threshold is set to 2, and $k = 8$ is used for the top-$k$ function. We follow the best model selection criteria from ZoomMIL: validation loss for CRC and CAMELYON16, and validation weighted-F1 for BRIGHT.

**vs. ZoomMIL.**   The most distinctive advantage of SpecMIL is its ability to sample relevant patches at low magnification using fully informed high-frequency features and geometric positional encoding, while eliminating the need for a differentiable top-k operation. This approach significantly reduces GPU usage, as shown in Fig 4, making it feasible for many downstream tasks, including end-to-end training. Additionally, contrary to the results reported in the ZoomMIL paper, where the non-differentiable top-k method led to reduced performance, our approach significantly outperforms ZoomMIL across all benchmark datasets. This demonstrates that if sufficient information, such as high-frequency features and geometric positional encoding, is provided at low magnification, it is possible to achieve meaningful improvements in the selection of high magnification patches.

**Ablation Study.**   To assess the impact of each component in our model, we performed a series of ablation studies, as summarized in Table 4. Reducing the number of selected patches $k$ to 300 leads to a decrease in performance, but the model still achieves comparable results due to the inclusion of all modules. Both the HF features and the LGPE module prove to be critical, as their inclusion leads to significant performance gains, highlighting their essential roles. The LGPE-MLP module, which leverages global positional encoding without utilizing the feature-based adjacency matrix $A_{\text{HF}}$, further demonstrates the careful design and effectiveness of the LGPE. The misaligned LGPE module,

| | Weighted-F1 (%) | Acc (%) |
|---|---|---|
| **SpecMIL** ($k = 600$) | **87.0** $\pm$1.3 | **87.5** $\pm$1.2 |
| w/ $k = 300$ | 84.6 $\pm$1.8 | 85.3 $\pm$1.6 |
| w/o HF | 80.0 $\pm$3.3 | 81.0 $\pm$2.8 |
| w/o LGPE | 80.5 $\pm$3.6 | 80.8 $\pm$3.2 |
| w/ LGPE-MLP (w/o $A_{\text{HF}}$) | 79.8 $\pm$4.4 | 80.4 $\pm$4.1 |
| w/ misaligned LGPE | 82.5 $\pm$4.7 | 83.2 $\pm$4.8 |

Table 4:   Ablation studies on the CAMELYON16 dataset. LGPE-MLP refers to the LGPE module that uses an MLP instead of a GNN, thereby eliminating geometry based on high-frequency features (local connectivity). Misaligned LGPE refers to the LGPE module where the indices for positional encoding $z^p$ are not aligned across different magnifications, such that $\bar{z}_{ij}^{PE} = [\bar{z}_{ij}; \bar{z}_{ij}^{h}; z_k^p]$ with a random index $k$ instead of $z_i^p$.

which assigns positional encoding $z_k^p$ with a random index $k$ at the high magnification, shows that while performance diminishes, it does not collapse entirely. This outcome is likely attributed to the fact that accurate positional encoding at lower magnifications still provides valuable information.

## 6   CONCLUSION

We have presented **Multi-Instance Learning with Spectral Methods** (SpecMIL), a spectral multiple-instance learning framework that resolves the long-standing multi-resolution dilemma in gigapixel image analysis. By extracting scale-resilient spectral features and encoding geometric relationships in graph-spectral features, SpecMIL identifies the most informative regions at low magnification and zooms in only where additional detail is essential. This strategy preserves high-frequency cues that conventional pipelines discard, while avoiding the prohibitive cost of exhaustively scanning every high-resolution patch. The proposed approach is conceptually simple, hardware-friendly, and broadly applicable to any domain where images far exceed the receptive field of modern vision encoders—including remote sensing, large-scale scene understanding, high-resolution microscopy, and digital pathology. We believe these findings open new avenues for scalable analysis of ever-growing visual data and lay the groundwork for future work in multi-resolution representation learning.

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

## A    FREQUENCY ANALYSIS FROM THE MAIN TEXT

To analyze the frequency components of Whole Slide Images (WSIs) with RGB channels, we apply a two-dimensional Fast Fourier Transform (2D FFT) to each color channel (R, G, B) of the image. The steps involved in obtaining and plotting the frequency spectrum (Figure 1 of the main text) are detailed below.

### A.1    2D FFT CALCULATION AND SHIFT

Given an RGB image $I(x, y)$ of size $N \times M$, where $x, y$ represent the spatial coordinates and each pixel has three values corresponding to the R, G, and B channels, the image can be represented as:

$$I(x, y) = [I_R(x, y), \quad I_G(x, y), \quad I_B(x, y)]$$

where $I_R(x, y)$, $I_G(x, y)$, and $I_B(x, y)$ represent the Red, Green, and Blue channels, respectively.

A 2D FFT is applied to each channel separately to transform the image from the spatial domain to the frequency domain. The 2D FFT for each channel is computed as:

$$F_C(u, v) = \sum_{x=0}^{N-1} \sum_{y=0}^{M-1} I_C(x, y) \cdot \exp\left(-i2\pi \left(\frac{ux}{N} + \frac{vy}{M}\right)\right)$$

where $C \in \{R, G, B\}$ denotes the color channel, $F_C(u, v)$ represents the frequency domain representation, and $u, v$ are the frequency coordinates.

The output of the 2D FFT has the low-frequency components located at the corners of the spectrum. To center the zero-frequency component (DC component), we apply a shift operation using:

$$F'_C(u, v) = \text{fftshift}(F_C(u, v))$$

where $F'_C(u, v)$ is the shifted spectrum with the DC component at the center.

### A.2    MAGNITUDE CALCULATION

For each color channel, the magnitude of the frequency components is computed as:

$$\text{Magnitude}_C(u, v) = \sqrt{\text{Re}(F'_C(u, v))^2 + \text{Im}(F'_C(u, v))^2}$$

where $\text{Re}(F'_C(u, v))$ and $\text{Im}(F'_C(u, v))$ are the real and imaginary parts of $F'_C(u, v)$, respectively.

### A.3    NORMALIZATION

To normalize the magnitude values, we divide each magnitude by the magnitude of the DC component:

$$\text{Normalized Magnitude}_C(u, v) = \frac{\text{Magnitude}_C(u, v)}{\text{Magnitude}_C(0, 0)}$$

This step ensures that the magnitude values are normalized relative to the DC component.

### A.4    FREQUENCY SPECTRUM

We compute the distance from the center (DC component) for each frequency component, which we consider as the frequency value:

$$d(u, v) = \sqrt{(u - u_0)^2 + (v - v_0)^2}$$

where $u_0 = \frac{N}{2}$ and $v_0 = \frac{M}{2}$ represent the center coordinates of the frequency spectrum.

## A.5 AVERAGING MAGNITUDES FOR EACH FREQUENCY

For each unique distance $d$, the magnitudes across all channels and all points equidistant from the center are averaged:

$$\text{Average Magnitude}(d) =$$
$$\frac{1}{N_d}\frac{1}{|C|}\sum_{\substack{(u,v)\\ \text{s.t. } d(u,v)=d}}\sum_{C}\text{Normalized Magnitude}_C(u,v)$$

where $N_d$ is the number of frequency components at distance $d$.

## A.6 PLOTTING THE FREQUENCY SPECTRUM

Finally, the frequency spectrum (Figure 1 in the main text) is plotted by setting the x-axis as the frequency distance $d$ and the y-axis as the corresponding averaged normalized magnitude Average Magnitude$(d)$. This plot represents the distribution of frequency components in the image.

# B RELATIONSHIP BETWEEN MAGNIFICATION AND FREQUENCY COMPONENTS IN WHOLE SLIDE IMAGES

In WSIs, observing tissue morphology at different magnifications reveals how the image's frequency components change. Specifically, an identical morphological feature viewed at high magnification (e.g., $20\times$) becomes a high-frequency component when viewed at a lower magnification (e.g., $10\times$). This section explains this phenomenon mathematically and underscores the significance of high-frequency components at lower magnifications in preserving morphological details.

## B.1 SPATIAL RESOLUTION AND FREQUENCY COMPONENTS

Consider an image $I(x, y)$ of size $N \times M$ pixels, representing tissue morphology at a high magnification $M_1$ (e.g., $20\times$). The corresponding frequency domain representation is obtained using a 2D Fourier transform:

$$F_{M_1}(u,v) = \sum_{x=0}^{N-1}\sum_{y=0}^{M-1} I(x,y) \cdot \exp\left(-i2\pi\left(\frac{ux}{N} + \frac{vy}{M}\right)\right)$$

At this high magnification, fine morphological details are distributed across a broader range of spatial frequencies, where lower frequencies represent coarse structures, and higher frequencies capture fine details.

When the same morphological feature is observed at a lower magnification $M_2$ (e.g., $10\times$) while maintaining the same image size $N \times M$, the spatial extent of the feature within the image is reduced. Since the image size remains constant, this reduction results in a denser packing of the morphology into the same pixel grid, effectively increasing the frequency of the corresponding components:

$$I_{M_2}(x,y) = I\left(\frac{x}{k}, \frac{y}{k}\right) \quad \text{with } k = \frac{M_1}{M_2}$$

where $k > 1$ represents the scaling factor between the magnifications.

By maintaining the same image size $N \times M$ at lower magnification $M_2$, the morphological features that were previously spread out at $M_1$ now occupy fewer pixels. This compression of spatial information into the smaller number of pixels results in an increase in the image's high-frequency components. Mathematically, this can be represented as:

$$F_{M_2}(u,v) = \sum_{x=0}^{N-1}\sum_{y=0}^{M-1} I_{M_2}(x,y) \cdot \exp\left(-i2\pi\left(\frac{ux}{N} + \frac{vy}{M}\right)\right)$$

Because the same morphological feature is now represented within a smaller spatial region (due to lower magnification), the rate of intensity change across pixels increases, leading to a shift in energy towards higher frequencies in the Fourier domain.

### B.2 Compression and High-Frequency Components

The compression of morphological features into a smaller space at lower magnification increases their associated frequency components. In particular, the Fourier transform of the lower magnification image $I_{M_2}(x, y)$ results in a frequency spectrum where the high-frequency components are more prominent compared to the higher magnification image $I(x, y)$. This can be mathematically understood as a redistribution of the image's energy towards higher frequencies:

$$F_{M_2}(u, v) = k^2 \cdot F_{M_1}(ku, kv)$$

This equation shows that when reducing magnification from $M_1$ to $M_2$, the frequency components $(u, v)$ are effectively scaled by the factor $k$, causing high-frequency components to dominate the spectrum.

### B.3 Importance of High-Frequency Components at Low Magnification

High-frequency components in the lower magnification image $M_2$ are critical because they retain information about the fine morphological details that would otherwise be visible only at higher magnifications. While the low-frequency components capture general shapes and structures, the high-frequency components contain details about edges, textures, and small-scale features that are essential for accurate interpretation of the tissue morphology.

These high-frequency components serve as vital clues to the underlying morphology at higher magnifications. For instance, subtle changes in tissue structure that are easily detectable at $20\times$ magnification manifest as high-frequency variations at $10\times$ magnification. Analyzing these high-frequency components at lower magnifications allows pathologists and automated systems to infer important morphological characteristics, potentially identifying regions of interest that warrant further investigation at higher magnification.

In summary, when maintaining the same image size (considering the consistent input size for a vision encoder) while lowering magnification, the resulting image inherently contains more high-frequency components due to the compression of morphological features. These components are essential for preserving and interpreting fine morphological details across different levels of magnification in WSIs.

## C High-Pass Filtering Process

High-pass filtering is a process used to remove or attenuate low-frequency components of an image while preserving or enhancing high-frequency components. This technique is particularly useful for edge detection, sharpening, and enhancing fine details in an image. The steps involved in applying a high-pass filter to a 256x256 RGB image using its 2D Fourier transform are outlined as follows:

### C.1 Frequency Domain Representation

Given the frequency spectrum $F_C(u, v)$ of each color channel $C \in \{R, G, B\}$ of the image, the frequency domain representation is shifted such that the DC component (low frequencies) is centered. This shifted representation is denoted as $F'_C(u, v)$.

### C.2 High-Pass Filter Design

A high-pass filter is designed by creating a filter matrix $H(u, v)$ of the same size as the frequency domain representation. The filter matrix is defined such that low-frequency components are attenuated (i.e., set to zero or reduced), and high-frequency components are preserved or passed through:

$$H(u, v) = \begin{cases} 0, & \text{if } d(u, v) < D_0 \\ 1, & \text{if } d(u, v) \geq D_0 \end{cases}$$

where $d(u, v) = \sqrt{(u - u_0)^2 + (v - v_0)^2}$ is the distance from the center (DC component), and $D_0$ is a threshold frequency that determines the cutoff between low and high frequencies. Frequencies with $d(u, v)$ less than $D_0$ are attenuated, while those greater than or equal to $D_0$ are retained.

## Original image $I$

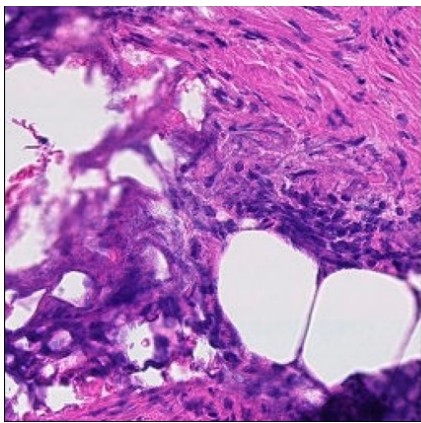

## Filtered image $I'$

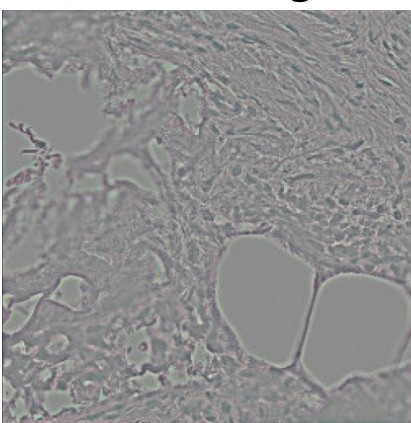

Figure 5: (Left) Original image and (right) high-frequency filtered image.

**Gaussian Filter.** A high-pass Gaussian filter is defined by inverting a standard Gaussian filter, which smoothly attenuates the low frequencies while preserving the high frequencies. This is mathematically expressed as:

$$H(u, v) = 1 - e^{-\frac{d(u,v)^2}{2\sigma^2}}$$

where $\sigma$ is the standard deviation that controls the spread of the Gaussian function.

In this filter, frequencies close to the center (low frequencies) are heavily attenuated, while higher frequencies are preserved or passed through. The parameter $\sigma$ determines the cutoff and transition between low and high frequencies; a smaller $\sigma$ results in a sharper transition, making the filter more selective towards high frequencies.

### C.3 APPLYING THE HIGH-PASS FILTER

The high-pass filter is applied to the frequency domain representation by element-wise multiplication:

$$F_C''(u, v) = F_C'(u, v) \cdot H(u, v)$$

where $F_C''(u, v)$ is the filtered frequency domain representation for each color channel.

### C.4 INVERSE FOURIER TRANSFORM

To convert the filtered frequency domain representation back to the spatial domain, an inverse 2D FFT is applied:

$$I_C'(x, y) = \text{Re}\left\{ \sum_{u=0}^{N-1} \sum_{v=0}^{M-1} F_C''(u, v) \cdot \exp\left( i2\pi \left( \frac{ux}{N} + \frac{vy}{M} \right) \right) \right\}$$

where $I_C'(x, y)$ is the high-pass filtered image in the spatial domain for each color channel, and $\text{Re}\{\cdot\}$ denotes taking the real part of the inverse transform.

### C.5 COMBINING THE CHANNELS

The final high-pass filtered image is obtained by combining the filtered channels:

$$I'(x, y) = [I_R'(x, y), \quad I_G'(x, y), \quad I_B'(x, y)]$$

where $I_R'(x, y)$, $I_G'(x, y)$, and $I_B'(x, y)$ are the high-pass filtered Red, Green, and Blue channels, respectively.

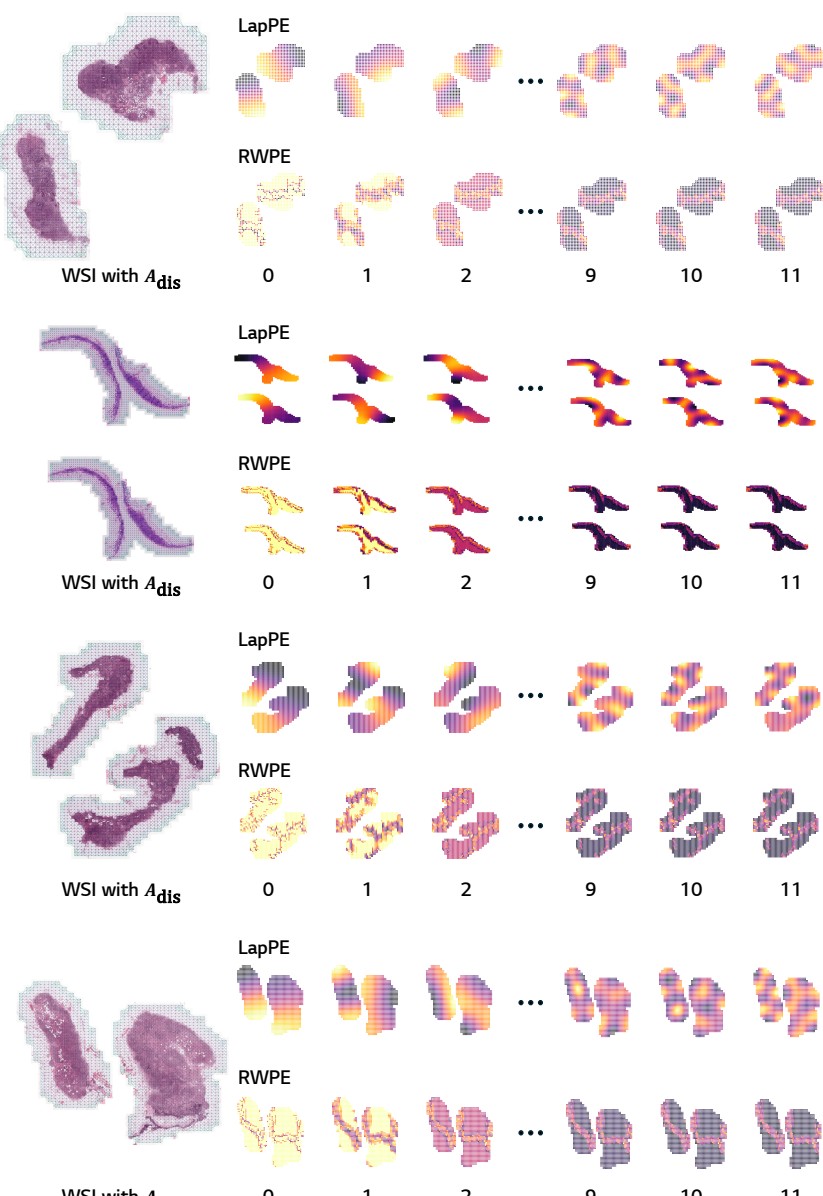

Figure 6: Visualization of two types of global positional encodings. LapPE and RWPE show the visualization of positional encoding corresponding to indices 0, 1, 2, ..., 9, 10, 11 of the graph Laplacian positional encoding and Random Walk, respectively. LapPE illustrates the transition from smooth, large-scale patterns at lower indices to more intricate, rapidly varying patterns at higher indices, capturing both global and local structural features of the graph. This pattern is analogous to the sinusoidal function commonly used as positional encoding in Euclidean space.

## C.6    RESULTING IMAGE

The resulting image $I'(x, y)$ emphasizes the high-frequency components, such as edges and fine details, while reducing or removing the low-frequency components. This effect can be visualized as a sharper, more detailed version of the original image, with less influence from gradual intensity changes and smooth regions.

# D GLOBAL POSITIONAL ENCODING

Global positional encoding helps to identify the relative position of each patch within irregular structures and to preserve structural information. By incorporating the spatial relationships between patches into a graph structure, positional encoding captures both local and global tissue architecture, enabling a more nuanced understanding of complex biological patterns. This approach preserves the geometric continuity and contextual information that are often lost when analyzing WSIs. Consequently, the model can better recognize patterns such as tumor boundaries, infiltrative growth, and other morphological features that rely on the spatial arrangement of tissue components, leading to more accurate and robust pathological assessments. We visualize two types of global positional encodings in Figure 6.

**Invariance to slide-level rotation.** A critical property of positional encoding in the context of WSIs is its invariance to slide-level rotations. Since WSIs typically lack a consistent orientation—meaning there is no predefined "top," "bottom," "left," or "right"—it is essential that the positional encoding scheme remains robust to such variations. This invariance ensures that the encoded positions of pixels or regions within the slide are not affected by arbitrary rotations of the image, which is a common occurrence in histopathological analysis. Consequently, the positional encoding must be designed in a manner that it provides a consistent and orientation-independent representation of spatial information. When constructing the global positional encoding from the distance-based adjacency matrix $A_{\text{dis}}$, it satisfies rotation invariance as demonstrated in the following proposition.

**Proposition 2** (Rotation Invariance of the Adjacency Matrix Derived from Euclidean Distances). *Let $P_1, P_2, \ldots, P_n$ be the coordinates of $n$ patches in a plane, and let $d(P_i, P_j)$ denote the Euclidean distance between patches $P_i$ and $P_j$. Define the adjacency matrix $A$ such that the $(i, j)$-th entry $A_{ij} = d(P_i, P_j)$. If the slide is rotated by an angle $\theta$ around a point $O$, then the resulting adjacency matrix $A'$ after rotation is identical to $A$, i.e., $A' = A$.*

*Proof.* Let the coordinates of patch $P_i$ be $(x_i, y_i)$ before rotation. The distance between two patches $P_i$ and $P_j$ is given by:

$$d(P_i, P_j) = \sqrt{(x_i - x_j)^2 + (y_i - y_j)^2}$$

After rotating the slide by an angle $\theta$ around a point $O$, the new coordinates $(x_i', y_i')$ of patch $P_i$ are given by the rotation matrix:

$$\begin{pmatrix} x_i' \\ y_i' \end{pmatrix} = \begin{pmatrix} \cos\theta & -\sin\theta \\ \sin\theta & \cos\theta \end{pmatrix} \begin{pmatrix} x_i \\ y_i \end{pmatrix}$$

This results in:

$$x_i' = x_i \cos\theta - y_i \sin\theta$$
$$y_i' = x_i \sin\theta + y_i \cos\theta$$

The distance between the rotated patches $P_i'$ and $P_j'$ is:

$$d(P_i', P_j') = \sqrt{(\delta x)^2 + (\delta y)^2}$$
$$\delta x = (x_i' - x_j'), \delta y = (y_i' - y_j')$$

Substituting the rotated coordinates:

$$(\delta x)^2 = [(x_i \cos\theta - y_i \sin\theta) - (x_j \cos\theta - y_j \sin\theta)]^2$$
$$= [(x_i - x_j) \cos\theta - (y_i - y_j) \sin\theta]^2$$
$$(\delta y)^2 = [(x_i \sin\theta + y_i \cos\theta) - (x_j \sin\theta + y_j \cos\theta)]^2$$
$$= [(x_i - x_j) \sin\theta + (y_i - y_j) \cos\theta]^2$$

Expanding and combining like terms:

$$(\delta x)^2 + (\delta y)^2 = (x_i - x_j)^2(\cos^2\theta + \sin^2\theta)$$
$$+ (y_i - y_j)^2(\cos^2\theta + \sin^2\theta)$$

Using the trigonometric identity $\cos^2\theta + \sin^2\theta = 1$:

$$d(P_i', P_j') = \sqrt{(x_i - x_j)^2 + (y_i - y_j)^2} = d(P_i, P_j)$$

Since $d(P_i', P_j') = d(P_i, P_j)$ for all $i, j$, the adjacency matrix $A'$ after rotation is identical to the original adjacency matrix $A$. Thus, the adjacency matrix is invariant under rotation. $\square$

## D.1 Graph Laplacian Positional Encoding

Given a distance-based adjacency matrix, we calculate the graph Laplacian and apply a spectral graph technique to extract eigenvectors Belkin & Niyogi (2003), which are used as global positional encodings.

1. **Construct the Adjacency Matrix $A$:** For a graph with $n$ nodes, the adjacency matrix $A$ is an $n \times n$ matrix where $A_{ij}$ represents the weight of the edge between node $i$ and node $j$. If there is no edge between them, $A_{ij} = 0$.

2. **Compute the Degree Matrix $D$:** The degree matrix $D$ is a diagonal matrix where each diagonal element $D_{ii}$ is the degree of node $i$, i.e., the sum of the weights of all edges connected to node $i$:

$$D_{ii} = \sum_j A_{ij}$$

3. **Calculate the Graph Laplacian $\Delta$:** The unnormalized graph Laplacian is computed as:

$$\Delta = D - A$$

Alternatively, the normalized graph Laplacian that we used can be computed as:

$$\Delta_{\text{sym}} = I - D^{-\frac{1}{2}} A D^{-\frac{1}{2}},$$

or:

$$\Delta_{\text{rw}} = I - D^{-1} A,$$

where here $I$ is the identity matrix.

4. **Compute the Eigenvectors of the Graph Laplacian:** Perform eigen decomposition of the Laplacian matrix $\Delta$:

$$\Delta = U^\top \Lambda U$$

where $\Lambda$ represents the eigenvalues and $U$ represents the corresponding eigenvectors. The eigenvectors corresponding to the smallest non-zero eigenvalues (excluding the zero eigenvalue, which corresponds to the all-ones vector) are used as the positional encodings.

5. **Select the Positional Encodings:** Choose the top $k_{PE}$ eigenvectors (excluding the first eigenvector if it corresponds to the zero eigenvalue) to form the positional encodings (i.e. $X_{PE} = U_{[:,:k_{PE}]}$). These eigenvectors provide a low-dimensional representation of the nodes' positions relative to the graph structure.

## D.2 Random Walk Positional Encoding

Random Walk is a method for initializing the positional representations of nodes in a graph Dwivedi et al. (2021). Random Walk Positional Encoding (RWPE) is based on the random walk diffusion process and serves as a $X_{PE}$ in the proposed Learnable Geometric Positional Encodings (LGPE) module.

### D.2.1 Definition of RWPE

RWPE is defined using the random walk operator, denoted as $RW = AD^{-1}$, where $A$ is the adjacency matrix of the graph, and $D$ is the degree matrix. The RWPE for a node $i$ is a vector that captures the probabilities of returning to node $i$ after $k$ steps of a random walk.

Formally, the RWPE for node $i$ is defined as:

$$X_{PE,i} = \left[ RW_{ii}, RW_{ii}^2, \ldots, RW_{ii}^k \right] \in \mathbb{R}^k$$

Here, $RW_{ii}^k$ represents the probability of returning to node $i$ after $k$ steps of the random walk.

# E    ADJACENCY MATRIX BASED ON HIGH-FREQUENCY FEATURES

As shown in Figure 7, high-frequency feature-based adjacency in the context of WSIs effectively captures the geometry and spatial arrangement of various tissue components, such as fat cells, metastatic cells, cytoplasm, and glandular structures. High-frequency features are sensitive to fine details, including edges, textures, and small-scale variations within the tissue, which are crucial for representing the distinct boundaries and complex geometries of these biological components. For instance, the well-defined boundaries of fat cells, the irregular structures introduced by metastasis, the textural variations within cytoplasm, and the intricate architecture of glandular structures all contribute to high-frequency signals. When a graph is constructed where patches are nodes and edges are based on high-frequency feature similarity, the resulting adjacency structure can accurately reflect the local geometric relationships between these components. This makes high-frequency feature-based adjacency particularly useful for mapping the tissue's microarchitecture and understanding the spatial organization of its constituent elements.

In $A_{HF}$ of LGPE, the edges between nodes are determined by feature-based similarity using a top-$k$ approach with distance thresholding. The nature of the features used—specifically, whether they are low-frequency or high-frequency—plays a critical role in determining the localization of adjacency within the graph.

| CRC | | | | | | | |
|---|---|---|---|---|---|---|---|
|  | ABMIL | CLAM-SB | TransMIL | WiKG | DSMIL | ZoomMIL | SpecMIL |
| **TFLOPs** | 13.63 | 13.63 | 13.63 | 13.63 | 17.94 | 1.06 | 2.24 |
| **Time (s)** | 4.85 | 4.85 | 4.85 | 4.85 | 6.37 | 0.38 | 0.41 |
| **BRIGHT** | | | | | | | |
|  | ABMIL | CLAM-SB | TransMIL | WiKG | DSMIL | ZoomMIL | SpecMIL |
| **TFLOPs** | 16.45 | 16.45 | 16.46 | 16.45 | 21.66 | 0.40 | 0.83 |
| **Time (s)** | 5.86 | 5.86 | 5.86 | 5.86 | 7.69 | 0.14 | 0.16 |
| **CAMELYON16** | | | | | | | |
|  | ABMIL | CLAM-SB | TransMIL | WiKG | DSMIL | ZoomMIL | SpecMIL |
| **TFLOPs** | 39.12 | 39.12 | 39.12 | 39.12 | 48.95 | 14.94 | 30.11 |
| **Time (s)** | 13.92 | 13.92 | 13.92 | 13.92 | 17.41 | 5.32 | 5.64 |

Table 5: Comparison of various models in terms of TFLOPs and Time (seconds).

## E.1    LOW-FREQUENCY FEATURE-BASED ADJACENCY

Low-frequency features capture broad, global structures in the image, such as large homogeneous regions, shapes, and general color distributions. When patches are connected based on low-frequency similarity, the graph reflects broader, more widespread similarities across the image, often linking spatially distant patches. As a result, the adjacency between patches is **less localized**, aligning with the widespread nature of low-frequency information.

## E.2    HIGH-FREQUENCY FEATURE-BASED ADJACENCY

Conversely, high-frequency features capture fine details within the image, such as edges, textures, and small-scale variations. These features vary more rapidly across spatial dimensions and are typically more specific to smaller, localized regions of the image. When graph edges are based on high-frequency feature similarity, the resulting adjacency is driven by more specific, localized characteristics. This means that patches need to be in closer proximity to share similar high-frequency features, leading to a graph structure where connections are primarily made between nearby patches. As a result, the adjacency in this graph is **more localized**, reflecting the detailed and localized nature of high-frequency information as visualized in Figure 8.

| | Resolution | CAMELYON16 | | BRIGHT | | CRC | |
|---|---|---|---|---|---|---|---|
| | | Weighted-F1 | Acc | Weighted-F1 | Acc | Weighted-F1 | Acc |
| MeanMIL | Low | 59.77 ±2.17 | 61.50 ±3.08 | 61.93 ±1.96 | 61.55 ±2.18 | 85.82 ±1.23 | 85.85 ±1.25 |
| | High | 60.03 ±2.02 | 62.27 ±2.93 | 63.61 ±4.26 | 63.40 ±4.36 | 87.32 ±0.88 | 87.32 ±0.93 |
| | Low + High (Naive) | 61.54 ±1.38 | 63.82 ±1.98 | 63.34 ±2.72 | 62.96 ±2.72 | 86.81 ±1.11 | 86.88 ±1.03 |
| MaxMIL | Low | 68.01 ±12.51 | 69.51 ±12.07 | 65.49 ±2.13 | 65.47 ±2.54 | 88.92 ±1.79 | 88.89 ±1.80 |
| | High | 68.85 ±9.04 | 74.68 ±3.64 | 64.92 ±3.92 | 65.47 ±4.57 | 87.93 ±1.90 | 87.81 ±2.07 |
| | Low + High (Naive) | 69.47 ±17.43 | 70.97 ±16.95 | 66.49 ±2.64 | 67.10 ±2.20 | 88.64 ±1.32 | 88.60 ±1.35 |
| ABMIL | Low | 76.00 ±1.84 | 76.92 ±1.77 | 65.88 ±2.83 | 65.58 ±2.27 | 88.85 ±0.93 | 88.84 ±0.96 |
| | High | 84.51 ±1.93 | 84.76 ±2.01 | 70.04 ±1.57 | 69.94 ±1.77 | 90.25 ±1.74 | 90.16 ±1.84 |
| | Low + High (Naive) | 82.24 ±2.13 | 82.52 ±2.06 | 68.46 ±2.70 | 68.63 ±3.50 | 90.34 ±1.05 | 90.36 ±1.11 |
| CLAM-SB | Low | 76.53 ±4.90 | 77.18 ±5.33 | 65.91 ±3.07 | 67.32 ±2.82 | 89.65 ±0.97 | 89.62 ±1.04 |
| | High | 83.34 ±3.06 | 83.64 ±3.01 | 68.01 ±3.20 | 68.30 ±3.57 | 91.60 ±1.33 | 91.58 ±1.35 |
| | Low + High (Naive) | 81.64 ±4.54 | 82.00 ±4.42 | 67.65 ±3.19 | 67.76 ±2.79 | 91.06 ±1.61 | 91.09 ±1.65 |
| TransMIL | Low | 75.16 ±3.86 | 75.71 ±4.25 | 62.94 ±3.96 | 64.05 ±3.95 | 88.94 ±2.84 | 88.89 ±2.88 |
| | High | 77.90 ±4.92 | 78.55 ±4.89 | 65.19 ±4.64 | 66.34 ±5.41 | 90.40 ±2.54 | 90.31 ±2.65 |
| | Low + High (Naive) | 74.68 ±3.50 | 76.23 ±3.36 | 61.15 ±5.45 | 61.87 ±5.79 | 87.88 ±2.87 | 87.91 ±2.61 |
| R²T-MIL | Low | 76.73 ±1.89 | 77.52 ±2.05 | 62.94 ±2.02 | 63.40 ±2.39 | 90.44 ±1.38 | 90.41 ±1.41 |
| | High | 80.25 ±5.05 | 80.62 ±5.19 | 69.64 ±2.18 | 69.83 ±2.89 | 91.44 ±0.91 | 91.43 ±0.91 |
| | Low + High (Naive) | 77.13 ±5.95 | 78.12 ±6.03 | 68.09 ±2.28 | 68.85 ±2.95 | 92.26 ±1.33 | 92.27 ±1.34 |
| WiKG | Low | 77.62 ±2.60 | 78.90 ±2.32 | 65.93 ±2.30 | 66.01 ±2.73 | 88.72 ±1.76 | 88.69 ±1.83 |
| | High | 86.92 ±2.68 | 87.25 ±2.72 | 68.76 ±2.01 | 68.95 ±2.86 | 91.93 ±1.39 | 91.92 ±1.43 |
| | Low + High (Naive) | 77.04 ±1.49 | 78.55 ±1.64 | 66.46 ±2.14 | 66.34 ±2.73 | 91.65 ±2.09 | 91.58 ±2.23 |

Table 6: Performance of MIL methods designed for single-resolution setting, including standard deviation values. The multi-resolution adaptation setting, denoted as *Low + High (Naive)*, represents the concatenation of features from different resolutions. These results underscore the significant challenges involved in utilizing multi-resolution approaches.

## F  IMPLEMENTATION DETAILS

For our experiments, we employed different learning rates depending on the dataset. Specifically, we used a learning rate of $2 \times 10^{-4}$ for the CAMELYON16 dataset and $1 \times 10^{-4}$ for the CRC and BRIGHT datasets. The Adam optimizer was utilized across all experiments, with a weight decay of $5 \times 10^{-4}$ for CAMELYON16 and no weight decay (0) for the CRC and BRIGHT datasets.

For positional encoding, we set $k_{PE} = 8$ for LapPE across all datasets. For the RWPE with concatenation, $k_{PE}$ was also set to 8, but this was only applied to the CAMELYON16 dataset. In the case of LGPE, we employed a single layer of SAGEConv Hamilton et al. (2017) for the Graph Neural Network (GNN). The hidden dimension was set to 256, with an output embedding dimension of 512, and the positional encoding dimension was $2 \times k_{PE}$. When constructing $A_{\text{dis}}$ and $A_{HF}$ with top-$k$, we select top 8 nodes (patches) and make them symmetric. For radius of high-frequency filter, we conducted a grid search over $\{10, 20, 30\}$ for the radius parameter and selected 10 as the optimal value.

We trained our models for 100 epochs, following the methodology outlined in the ZoomMIL paper Thandiackal et al. (2022). However, unlike ZoomMIL, which utilizes a single train/validation/test split, we conducted our experiments using three distinct splits to ensure robust experimental results.

Thandiackal et al. also highlight that the computational cost of the MIL modules is minimal compared to the dominant overhead introduced by patch feature extraction. Consequently, differences in FLOPs and processing time across methods are negligible and typically only detectable at very fine decimal levels. Our method follows a similar patch processing strategy (from low to high magnification) and requires a comparable number of high-magnification patches as ZoomMIL. While our approach introduces slightly higher computational demands due to the incorporation of HF features, this overhead is mitigated by efficient GPU parallelization. This allows for substantially larger batch sizes compared to ZoomMIL. In Table 5, we present the computational metrics on the CRC dataset.

We also present extended results in Table 6, including the multi-resolution configurations for baseline methods originally designed for single-resolution analysis. This includes traditional MIL approaches with instance-level pooling mechanisms, such as max-pooling (MaxMIL) and mean-pooling (Mean-MIL) Campanella et al. (2019), implemented through instance concatenation (i.e. patch-level concatenation). Additionally, we evaluate widely adopted MIL models for WSI analysis, including ABMIL Ilse et al. (2018), CLAM-SB Lu et al. (2021), and TransMIL Shao et al. (2021), using bag-level concatenation (i.e. slide-level concatenation). To further demonstrate the robustness and adaptability of our method, we validate its performance in combination with more recent approaches, such as R$^2$T-MIL Tang et al. (2024) and WiKG Li et al. (2024), also utilizing bag concatenation.

## G  PROCESS THREE MAGNIFICATIONS

Our method is capable of handling three magnifications, just like ZoomMIL. However, even when using their framework, we found ZoomMIL's performance with three magnifications to be non-reproducible. For instance, on the CAMELYON16 dataset, ZoomMIL achieved an F1 score of $68.0 \pm 2.8$ and accuracy of $69.9 \pm 3.2$. In contrast, our proposed method, SpecMIL, achieved significantly better results with an F1 score of $87.3 \pm 1.2$ and accuracy of $87.8 \pm 1.1$.

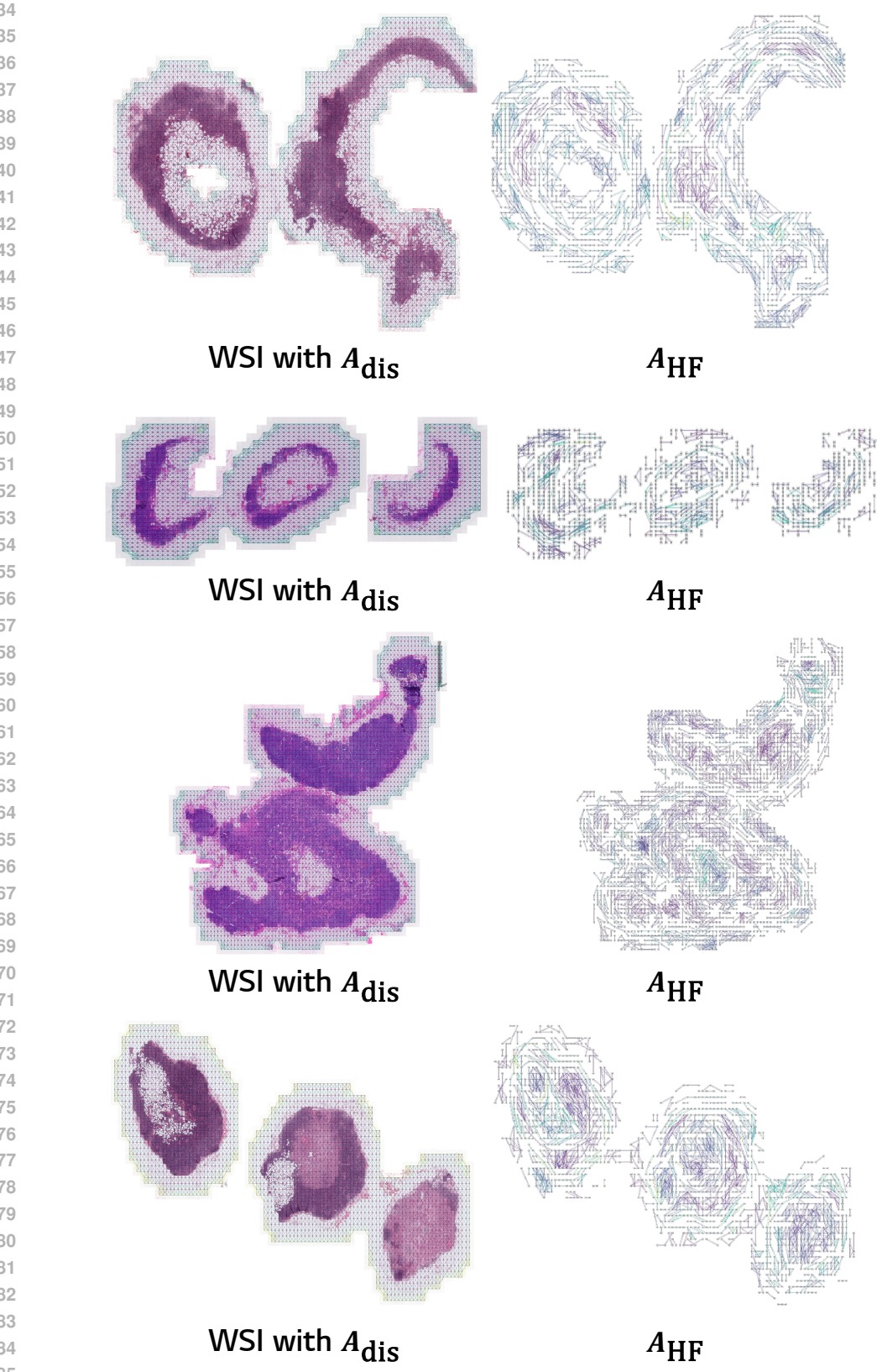

Figure 7: (Left) WSI with distance-based adjacency $A_{\text{dis}}$ and (right) visualization of $A_{HF}$. While $A_{\text{dis}}$ provides global connectivity, $A_{HF}$ encodes inherent geometry with localized patterns.

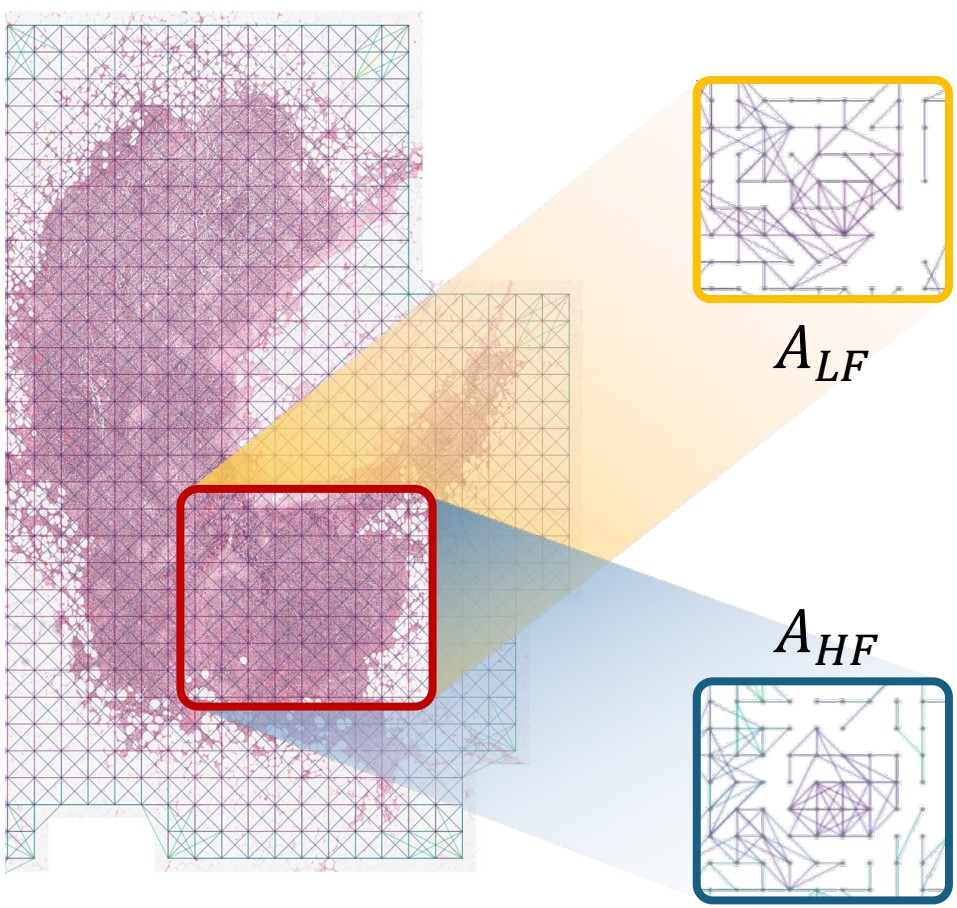

Figure 8: (Left) WSI with a distance-based adjacency $A_{\text{dis}}$ and (right) comparison of feature-based adjacency between low-frequency ($A_{LF}$) and high-frequency ($A_{HF}$). $A_{HF}$ exhibits more localized connectivity.

