# OpenReview forum: "Spectral Multiple-Instance Learning for Efficient Gigapixel Image Analysis"
_ICLR.cc/2026/Conference — ICLR 2026 Conference Withdrawn Submission_

### Official Review · Reviewer_tNbV · 2025-10-30

**Soundness:** 3
**Presentation:** 3
**Contribution:** 3
**Rating:** 4
**Confidence:** 3

**Summary:**

This paper proposes SpecMIL, a spectral multiple-instance learning framework that extracts informative high-frequency cues at low magnification, encodes cross-patch geometric relationships via spectral graph methods, and selectively “zooms in” only on high-value regions, thereby avoiding exhaustive high-magnification processing while retaining discriminative details. Key components include spectral high-frequency encoding on low-magnification patches; a learnable geometric positional encoding (LGPE) constructed from distance- and feature-based graphs with rotation-invariance; an LGPE-aware gated attention (LPGA) aggregator; and multi-scale selective zooming with cross-scale feature fusion. On CAMELYON16, BRIGHT, and CRC WSI benchmarks, the authors report SpecMIL achieves competitive or superior performance compared to strong MIL baselines while reducing memory usage (e.g., ~20% less GPU memory than ZoomMIL on CAMELYON16).

**Strengths:**

The main strengths lie in the WSI-relevant priors and the practical design. Using low-magnification spectral cues to triage zoom decisions mirrors pathologist workflow and yields measurable efficiency gains without sacrificing performance. The rotation-invariant positional encoding is theoretically grounded and addresses scanner-induced orientation variability. The empirical coverage across standard benchmarks with informative ablations adds credibility, and the selective-zoom design likely cuts disk I/O by limiting high-magnification tile retrieval, which is often a hidden bottleneck in WSI systems.

**Weaknesses:**

The reliance on an ImageNet-pretrained ResNet‑50 backbone is out of step with current practice in WSI, where pathology-specific foundation encoders (e.g., CONCH [1]/UNI [2]) provide substantially stronger representations. Without benchmarking SpecMIL on top of these stronger features, it is unclear whether the reported gains persist once the encoder bottleneck is removed. This makes the comparative positioning less convincing.

Statistical rigor is underreported. The main results appear to be single-seed. Given slide-level heterogeneity and class imbalance, variance reporting across multiple random seed ablations is needed to substantiate claims.

Localization and interpretability are also underdeveloped: overlays of selected zoom regions and attention maps against available lesion annotations, with lesion-level metrics (e.g., FROC on CAMELYON lesions), would better demonstrate clinical utility.

[1] Chen, Richard J., et al. "Towards a general-purpose foundation model for computational pathology." Nature medicine 30.3 (2024): 850-862.
[2] Lu, Ming Y., et al. "A visual-language foundation model for computational pathology." Nature medicine 30.3 (2024): 863-874.

**Questions:**

Have you evaluated SpecMIL with pathology foundation features (e.g., CONCH, UNI)? If so, does SpecMIL still provide accuracy and efficiency gains, and what changes in zoom selection behavior do you observe? If not, what constraints prevented this evaluation?

Could you provide interpretability evidence with qualitative overlays and quantitative localization metrics? For datasets with lesion annotations, show the selected zoom regions and attention heatmaps aligned with ground-truth masks, and report lesion-level metrics to substantiate the claim that SpecMIL “zooms” to diagnostically relevant areas.

---

### Official Review · Reviewer_3osZ · 2025-11-01

**Soundness:** 3
**Presentation:** 2
**Contribution:** 2
**Rating:** 2
**Confidence:** 4

**Summary:**

The authors introduce SpecMIL, a spectral MIL framework for gigapixel image analysis. It mitigates the multi-resolution challenge in WSI by extracting high-frequency cues from low-magnification patches and using graph spectral structure to guide targeted high-resolution zoom-in. Results on multiple pathology benchmarks (e.g., tumor subtyping, grading, and metastasis detection) show improved performance.

**Strengths:**

- Originality: The author proposed a method that combine theories of graph and signal processing to mitigate the multi-scale issue in pathology
- Soundness of methodology: The author provide sufficient theoretical justification and examples of the proposed framework

**Weaknesses:**

- The experiment is not sufficient to support the hypothesis and the theory: The main narrative of the paper is that: Digital pathology processing suffer from multi-scale issue-> a theory grounded method proposed to mitigate so that we ideally should see better performance since a key issue is mitigated. However, the experiments show kind of contradictory. If we take the second best model from each task (Take Weighted F1 for example):
  - CAMELYON16: 86.9 ±2.7 (WiKG (20x)) vs 87.0 ±1.3 (SpecMIL)
  - BRIGHT: 70.0 ±1.6 (ABMIL (2.5x)) vs 70.7 ±4.7 (SpecMIL)
  - CRC: 92.3 ±1.3 (R2T-MIL (5× + 10×))  vs 92.3 ±0.8 (SpecMIL)

We can conclude that gap between other baseline without addressing multi-scale issue and SpecMIL is very small and single resolution is enough to get comparable results, which undermine the core narrative in the paper. Similar trend can also be observed on Acc. metric.
- The visualization is confusing: It's unclear what is visualized in Figure 3,7,8. it would be nice to show what the underlining tissue look like instead of just throwing bunch of connections and colors.

**Questions:**

Do the authors have any comment about why there is no clear advantages compared to the baselines without addressing multi-resolution issue?

---

### Official Review · Reviewer_CoCm · 2025-11-01

**Soundness:** 3
**Presentation:** 2
**Contribution:** 2
**Rating:** 4
**Confidence:** 4

**Summary:**

The paper introduces SpecMIL, a spectral-based multiple instance learning framework designed to address the multi-resolution challenge in gigapixel image analysis. By leveraging graph spectral representations of low-magnification images, the method aims to capture high-frequency structural information that guides selective zooming into diagnostically relevant regions. This approach seeks to balance global contextual understanding with efficient local detail exploration, reducing computational cost while maintaining high diagnostic accuracy.

**Strengths:**

1.The paper proposes a novel idea of introducing spectral graph analysis into the MIL framework to address the multi-resolution dilemma in gigapixel image analysis.
2.The approach is theoretically inspired by frequency-domain analysis, offering a new angle distinct from conventional attention- or transformer-based multi-scale learning methods.
3.Experimental validation appears to involve multiple pathology datasets and diverse tasks, which helps demonstrate general applicability.

**Weaknesses:**

1.The literature review on multiple instance learning appears outdated, with most references stopping around 2022. This raises concerns about whether the authors have adequately reviewed the most recent works and included up-to-date SOTA baselines for comparison.
2.The main comparison method, ZoomMIL, is capable of handling more than two magnifications. The proposed approach appears restricted to only two, which may limit its flexibility in modeling complex multi-scale relationships.
3.From Table 4, the hyperparameter k is evaluated with only two values (300 and 600), which is insufficient to convincingly demonstrate that 600 is the optimal choice.
4.A key difference from ZoomMIL is the use of frequency analysis to extract high-frequency components. However, the paper lacks sufficient evidence to show the effectiveness or benefits of this approach. Providing supporting analysis or visualizations would strengthen the work. The paper would benefit from a visualization showing that the regions highlighted by the method at low resolution correspond to diagnostically relevant structures. This would help confirm that the model focuses on meaningful features rather than high-frequency artifacts or spurious patterns.

**Questions:**

1.The sections “High-Frequency Components in WSIs” and “Learnable Structural and Positional Encoding” contain very few references. Could the authors clarify whether this is due to a lack of prior work in these areas, or are there relevant studies that could be cited to support these discussions?
2.What’s the performance of “w/o original features”?
3.Is the feature extractor a ResNet-50 pre-trained on natural images? Since pathology foundation models like UNI and Virchow provide higher-quality features for histopathology, could the authors explain why ResNet-50 was chosen instead?

---

### Official Review · Reviewer_V8Jh · 2025-11-03

**Soundness:** 2
**Presentation:** 2
**Contribution:** 2
**Rating:** 4
**Confidence:** 5

**Summary:**

This paper introduces SpecMIL, a novel multi-instance learning framework designed to tackle the multi-resolution dilemma in gigapixel image analysis. The core contributions are twofold: 1) a spectral high-frequency encoding method that uses Fourier analysis to identify diagnostically critical details at low magnification, and 2) a Learnable Geometric Position Encoding (LGPE) that captures rotation-invariant spatial relationships using graph spectral theory. By intelligently guiding a "zoom-in" process towards the most informative regions, SpecMIL achieves state-of-the-art performance on multiple whole-slide image benchmarks.

**Strengths:**

The primary strength of this paper is its novel and intuitive core idea. The strategy of using high-frequency features to identify regions of interest (ROIs) at low magnification is a clever and well-justified approach to the multi-resolution problem in gigapixel images. This method provides a more principled way to guide the "zoom-in" process compared to relying solely on spatial-domain features, which may become ambiguous at lower resolutions.

**Weaknesses:**

**1. Insufficient Analysis of Hyperparameter Sensitivity:** The method's performance appears to hinge on several crucial hyperparameters, yet their sensitivity is not adequately explored.
   - **High-Pass Filter Radius (`r`):** The choice of `r=10` is presented as optimal based on a grid search, but the paper lacks an analysis of how performance varies with different `r` values. This parameter fundamentally defines what constitutes a "high-frequency feature." A small `r` might retain noise, while a large `r` could discard subtle but diagnostically relevant details. A sensitivity plot showing F1-score as a function of `r` on a validation set would be essential to demonstrate the robustness of this choice and provide guidance for future applications.
   - **Top-K Selection (`K`):** The number of selected high-magnification patches, `K`, directly trades off performance and computational cost. The ablation study compares `K=600` and `K=300`, but a more comprehensive analysis is needed. How does performance saturate or degrade as `K` approaches the limits (e.g., very small or very large values)? Understanding this trade-off is critical for deploying the model in resource-constrained environments.

**2. Limited Scope of Experimental Validation:** While the experiments on three classification datasets are strong, the paper's claims of broad applicability would be more convincing with a wider range of tasks and domains.
   - **Lack of Segmentation Task Evaluation:** The core hypothesis of SpecMIL is identifying information-rich regions. This capability is arguably even more critical for pixel-level tasks like tumor segmentation than for slide-level classification. Evaluating SpecMIL's zooming strategy as a pre-processing or attention-guiding step for a segmentation model (e.g., on the Gleason 2019 or BCSS datasets) would provide powerful evidence of its utility beyond classification. Without this, the claim of resolving the "multi-resolution dilemma" is only partially substantiated.
   - **Unsupported Generalization Claims:** The paper suggests applicability to remote sensing and large-scale scene understanding without empirical support. These domains possess different feature characteristics; for instance, high-frequency signals in satellite imagery might correspond to benign textures (e.g., forests, water ripples) rather than objects of interest. Acknowledging these potential challenges and discussing necessary adaptations would make the claims more credible.

**3. Need for Deeper Mechanistic Insight through Visualization:** The paper successfully demonstrates *that* the method works via ablation studies, but provides limited insight into *why* it works.
   - **Qualitative Case Studies:** The most compelling evidence would be a qualitative analysis of selected patches. The authors should visualize a few concrete examples where a low-magnification patch appears innocuous to the naked eye (and perhaps to a baseline model without HFE) but is selected by SpecMIL due to strong high-frequency signals, and then show that the corresponding high-magnification view indeed contains critical diagnostic features (e.g., mitotic figures, micro-metastases). This would bridge the gap between the abstract concept of "high frequency" and concrete pathological practice.
   - **Failure Mode Analysis:** A discussion of failure cases is conspicuously absent. Are there specific tissue architectures or disease patterns (e.g., diffuse, non-localized tumors vs. solid, well-defined ones) where SpecMIL struggles? Understanding the method's limitations is as important as understanding its strengths.

**4. Ambiguity in the Delineation of Novelty:** The paper could do a better job of situating its contributions within the broader literature to clarify the precise boundaries of its novelty.
   - **Context of Fourier Features in DL:** While the application of Fourier analysis to guide multi-scale sampling is novel, the use of frequency-domain features in deep learning is not. For example, works like [Wang et al., "High-Frequency Component Helps Explain the Generalization of Convolutional Neural Networks," CVPR 2020] have already explored the role of frequency components. The authors should explicitly differentiate their work by highlighting how they move beyond analysis to create an *actionable signal* for a dynamic pipeline, which is a key distinction.
   - **Positioning of LGPE:** The Learnable Geometric Position Encoding (LGPE) combines established techniques (graph Laplacian eigenvectors, feature-based k-NN graphs). The novelty lies in the specific combination and its application. To strengthen this claim, a brief comparative discussion against other sophisticated graph positional encoding schemes (e.g., those in Graphormer [Ying et al., 2021]) would be beneficial, explaining why this particular simpler design is sufficient or even superior for the WSI domain.

**5. Incomplete Discussion of Computational Overhead:** The analysis of computational cost is somewhat superficial and could lead to a misunderstanding of the true overhead.
   - **Pre-processing Cost:** The reported timing results in Table 5 appear to cover only the model's forward/backward pass. However, the HFE module requires a significant one-time pre-processing step (FFT -> filter -> IFFT) for every single low-magnification patch. For a WSI containing hundreds of thousands of patches, this cost could be substantial. The authors should quantify this pre-processing time and storage overhead to provide a complete picture of the method's efficiency profile.

**Questions:**

1.  **Sensitivity to the high-pass filter radius `r`:** How sensitive is the model's performance to the choice of the radius `r`? Could you provide a brief analysis or a plot showing performance (e.g., F1-score) as `r` varies? This would help in understanding the robustness of the high-frequency feature extraction.

2.  **Qualitative examples of selected patches:** Could you provide a qualitative example comparing a patch selected by SpecMIL (due to its high-frequency content) with one that was not? Showing the low-magnification view, its high-frequency visualization, and the corresponding high-magnification ground truth would offer strong intuition for why the method is effective.

3.  **Clarification on pre-processing overhead:** What is the computational and time overhead of the initial high-frequency feature extraction (the FFT -> filter -> IFFT -> encode pipeline) for an entire WSI? A rough estimate would be helpful to assess the practical deployment cost of the method.

4.  **Applicability to segmentation tasks:** Have you considered how this zoom-in strategy might be adapted for dense prediction tasks like tumor segmentation? A brief discussion on its potential or challenges in that context would be valuable.

**Details Of Ethics Concerns:**

N/A.

---

### Note · Authors · 2025-12-03

**Comment:**

We would like to express our sincere gratitude to the AC and the anonymous reviewers for their careful evaluation and insightful comments on our manuscript. Although we have decided to withdraw the paper at this time, the reviewers’ suggestions have been extremely helpful and will significantly guide our future work on this topic.

**Withdrawal Confirmation:**

I have read and agree with the venue's withdrawal policy on behalf of myself and my co-authors.